# Perioperative Administration of Cystine and Theanine Suppresses Inflammation and Facilitates Early Rehabilitation and Recovery after Esophagectomy: A Randomized, Double-Blind, Controlled Clinical Trial

**DOI:** 10.3390/nu14112319

**Published:** 2022-05-31

**Authors:** Hiroshi Okamoto, Yusuke Taniyama, Tadashi Sakurai, Gaku Kodama, Chiaki Sato, Toshiaki Fukutomi, Yohei Ozawa, Hirotaka Ishida, Ken Koseki, Takuro Yamauchi, Toru Nakano, Michiaki Unno, Takashi Kamei

**Affiliations:** 1Department of Surgery, Tohoku University Graduate School of Medicine, 1-1, Seiryo-machi, Aoba-ku, Sendai 980-8574, Japan; yusuketaniyama@surg.med.tohoku.ac.jp (Y.T.); schiaki@surg.med.tohoku.ac.jp (C.S.); t-fukutomi@surg.med.tohoku.ac.jp (T.F.); yohei.ozawa@surg.med.tohoku.ac.jp (Y.O.); h-ishida@surg.med.tohoku.ac.jp (H.I.); kenko819@surg.med.tohoku.ac.jp (K.K.); taku.y.xc5@surg.med.tohoku.ac.jp (T.Y.); m_unno@surg.med.tohoku.ac.jp (M.U.); tkamei@surg.med.tohoku.ac.jp (T.K.); 2Izumi Nomura Family Clinic, 1-5, Nomura Katsurashima-Higashi, Izumi-ku, Sendai 981-3124, Japan; tsakurai@surg.med.tohoku.ac.jp; 3Department of Rehabilitation, Tohoku University Hospital, 1-1, Seiryo-machi, Aoba-ku, Sendai 980-8574, Japan; gaku_kodama@yahoo.co.jp; 4Department of Gastroenterologic and Hepato-Biliary-Pancreatic Surgery, Tohoku Medical and Pharmaceutical University, 1-12-1, Fukumuro, Miyagino-ku, Sendai 983-8512, Japan; torun@med.tohoku.ac.jp

**Keywords:** cystine, theanine, esophagectomy, inflammation, rehabilitation

## Abstract

Oral administration of cystine and theanine (CT) increases glutathione levels to modulate the inflammatory response, which has yet to be sufficiently explored for patients’ recovery and early rehabilitation. We planned a randomized, double-blind, placebo-controlled trial to determine whether perioperative oral administration of CT promotes recovery after esophagectomy. Patients were randomized into either CT or placebo groups, who received preoperative and postoperative treatments for 4 and 13 days, respectively. The main outcome measures were triaxial accelerometer readings, inflammation indicators, a 6 min walk test (6MWT), and a quality of life questionnaire (QoR-40). The study involved 32 patients. Although the CT group (*n* = 16) showed better patient activity across the investigated period, there was no significant difference between the two groups. However, white blood cell count on postoperative days (POD) 2 and 10, neutrophil count (POD 2, 7, and 10), and C-reactive protein level (POD 13) in the CT group were significantly lower than in the placebo group. Furthermore, 6MWT on POD 7 and QoR-40 on POD 13 were significantly higher in the CT group than those in the placebo group. This study suggests that perioperative administration of CT may contribute to early recovery and rehabilitation after esophagectomy via suppression of inflammatory response.

## 1. Introduction

Although esophagectomy is still highly invasive and has many associated complications [1], recent improvements in perioperative management targeted at an early recovery after surgery are still evolving [2]. In the context of patients’ satisfaction, it is important to establish an evidence-based management method and to provide information for patients with the following overarching goals: (1) to modulate biological responses to surgical insults; (2) to ensure early restoration of physical activity; (3) to gain early recovery of normal nutritional intake; and (4) to mitigate perioperative anxiety and to enhance motivation to recover, as advocated in ESSENSE (essential strategy for early normalization after surgery with patient’s excellent satisfaction) protocol [3]. Despite the reported progress and efforts during the perioperative period, there are still reported cases of delays in postoperative recovery, largely due to the stress caused by esophagectomy. Thus, a treatment approach that can mitigate stress against surgical invasion is required.

Cystine is a sulfur-containing amino acid made of two disulfide-bonded cysteines and is known as a precursor to glutathione (GSH). Theanine (gamma-glutamylethylamide) is an amino acid that is absorbed in the small intestine and hydrolyzed in the intestine and liver to glutamic acid and ethylamine. GSH is a tripeptide consisting of glutamic acid, cysteine, and glycine. Basic and animal model studies have shown that the oral administration of both cystine and theanine (CT) increases tissue GSH levels, reduces inflammatory responses, regulates immune responses, and reduces the biological response to surgical invasion [4,5,6].

A reduction in biological reactions from surgical invasion is necessary to promote postoperative recovery. Previous studies have proven that GSH concentration in blood and organs decreases during biological invasion [7,8]. GSH is an antioxidant derived from living tissue and is meant to protect the body from oxidative stress. It has been reported that GSH in the intracellular redox state can suppress the production of inflammatory cytokines and promote the function of immune cells [9]. CT are important supplements that are very crucial for the synthesis of GSH. Although the perioperative oral administration of CT reportedly alleviates postgastrectomy inflammation [10], such a finding is still scanty for esophagectomy. It is also unclear whether supplements contribute to the promotion of postoperative recovery. Therefore, we planned a randomized, double-blind, placebo-controlled trial to determine whether the perioperative oral administration of CT promotes recovery and facilitates early rehabilitation after esophagectomy in patients with esophageal cancer.

## 2. Materials and Methods

### 2.1. Trial Design and Randomization

This study was designed as a randomized double-blind placebo-controlled clinical trial. All eligible patients were randomized to receive either CT or a matching placebo, with a 1:1 allocation ratio between the two groups using a block randomization model.

### 2.2. Participants

#### 2.2.1. Inclusion and Exclusion Criteria

Patients who satisfied the following criteria were included: patients within the age range of 20–80 years at the time of registration; those who met the Eastern Cooperative Oncology Group criteria [11] with a Performance Status score of 0–1; patients booked for thoracoscopic esophagectomy performed in the prone position using a gastric tube only with or without preoperative treatment; those who met the tumor-node-metastasis classification (UICC classification 7th edition) for clinical stages I, II, or III, excluding T4 [12]; and patients whose written informed consent to participate in the study were independently obtained.

Patients with the following conditions were, however, excluded from the study: active infection; uncontrolled hypertension and diabetes mellitus; clinically problematic heart disease (e.g., ischemic heart disease or congestive heart failure); severe lung disease (e.g., interstitial pneumonia, pulmonary fibrosis, or severe emphysema); pregnancy and postpartum period within 28 days after delivery, or breastfeeding; or history of clinically problematic mental disorder or central nervous system disorder. Additionally, patients who were participating in other clinical trials at the time of this study, those who could not ingest orally, those currently under infusion of amino acid solution in the perioperative period, those who developed severe complications during the study, and other patients who were adjudged inappropriate as participants for this study by the principal investigator were excluded.

#### 2.2.2. Sample Size Calculation

It was calculated that 20 cases in each group were required to provide a significant difference with a two-sided significance level of 0.05 for a 0.80 power for testing the research hypothesis that the postoperative recovery rate of the CT group is significantly higher than that of the placebo group, with efficacy rates of 74% in the CT group and 30% in the placebo group.

### 2.3. Outcome Measures

#### 2.3.1. Metabolic Equivalents and Exercise

Metabolic equivalent (MET) is one of the major parameters of physical activity intensity; 1 MET is approximately 1 kcal kg h^−1^ body weight. It is used to quantify the activity characterized by sedentary behavior (1.0–1.5 METs), light intensity (1.6–2.9 METs), moderate intensity (3–5.9 METs), and vigorous intensity (≥6 METs) [13]. It has been widely used in several fields, including rehabilitation [14]. Exercise (Ex) is the value obtained by multiplying MET and the activity time (hour) when the physical activity intensity is 3 METs or more.

#### 2.3.2. Primary Endpoint

The primary endpoint was physical activity (exercise and number of steps per day) measured using a triaxial accelerometer (Active style Pro HJA-750C, OMRON HEALTHCARE, Co., Ltd., Kyoto, Japan) at 40 × 52 × 12 mm and 23 g including batteries. The dynamic range of each axis is from 3 mG to 6 G. The device measures triaxis acceleration data in 10 s epochs and calculates combined acceleration. Furthermore, living activity, walking activity, and METs are estimated from the combined acceleration [15,16]. Patients hooked the device onto their waist when they woke up and took it off during sleeping and bathing. Exercise and the number of steps were measured from 4 days preoperatively to 13 days postoperatively, excluding the day of the surgery. The recorded data (>480 min per day) in the accelerometer were used for the analysis [17].

#### 2.3.3. Secondary Endpoint

The six-minute walk test (6MWT) [18] was measured on the penultimate day of the surgery, POD 7, and POD 13 by an independent assessor (a physiotherapist). The distance walked by a patient within 6 min was measured. This test was performed to evaluate exercise tolerance.

White blood cell count, neutrophil count, lymphocyte count, C-reactive protein (CRP), serum albumin, total cholesterol, liver function, renal function, electrolytes, prealbumin, and retinol-binding protein (RBP) from the blood were also measured on the penultimate day of the surgery and POD 1–7, 10, and 13. The CONUT (controlling nutritional status) score is a well-established scale for estimating patients’ nutritional state, and it was calculated using albumin, lymphocyte count, and total cholesterol [19].We used the quality of recovery score (QoR-40) as a quality of life (QOL) questionnaire [20,21]. The questionnaire consists of 40 questions on the following subscales: Comfort, Emotions, Physical Independence, Patient Support, and Pain. Each item is rated on a five-point Likert scale. The patients completed the scale on the penultimate day of the surgery, POD 3, and POD 13. The tool was reported as a good objective measure of QoR after anesthesia and surgery [20] and had been used in various fields [22,23,24]. The QoR-40 has been shown to be superior among the 12 instruments in a meta-analysis for assessing QOL, and it was recommended for validation and application studies on short-term postoperative recovery [25]. Another quantitative systematic review included 3459 patients from 17 studies in nine countries to evaluate the validity, reliability, responsiveness, and clinical utility of QoR-40 and demonstrated that QoR-40 was a suitable measure of postoperative quality of recovery in a range of clinical and research situations [22].

#### 2.3.4. Adverse Events

Any adverse events were determined according to The National Cancer Institute Common Terminology Criteria for Adverse Events ver. 4.0 [26].

### 2.4. Study Procedure

The study protocol and consent form were approved by the ethical committee of Tohoku University Hospital (accession number 2016-2-151), and informed consent was obtained from all patients. This study was registered at the University Hospital Medical Information Network clinical trials registry in Japan on 16 November 2016 (registration number: UMIN000024849). The procedure complied with the guidelines of the Declaration of Helsinki [27].

#### 2.4.1. Supplement and Placebo

The dosages of cystine (700 mg) and theanine (280 mg) were determined in conformity with those used in previous studies [4,10]. The test food used for CT, in stick form, consisted of 700 mg cystine, 280 mg theanine, 15 mg citric acid, 1.5 mg aspartame, and 500 mg maltitol (1.5 g, 5.0 kcal). The placebo, in a similar stick form, consisted of 880 mg of isomaltulose (palatinose), 98 mg of powder syrup, 15 mg of citric acid, 1.1 mg of aspartame, and 500 mg of maltitol (1.3 g, 4.3 kcal). Either the CT or the placebo was administered to the patients 4 days preoperatively and 13 days postoperatively, excluding the day of the surgery. The drug was administered through a feeding tube and jejunostomy after surgery. It was administered orally when the patients became conscious and stable. Ajinomoto Co, Inc. (Tokyo, Japan) provided the test foods and lent us accelerometers for free.

#### 2.4.2. Surgical Procedure

Methylprednisolone sodium succinate (250 mg per body) was intravenously infused into the patients before surgery. Thoracoscopic esophagectomy in the prone position, and hand-assisted laparoscopic or laparotomy gastric tube reconstruction via posterior mediastinal route were performed. Three-field lymph node dissection was also performed for tumors of the upper or middle thirds of the esophagus; two-field lymph node dissection was performed for tumors of the lower thirds of the esophagus [28].

### 2.5. Data Analysis

All statistical analyses were performed using JMP Pro Version 15 (SAS Institute Japan, Tokyo, Japan). Continuous data were assessed using Student’s t-test, while categorical data were analyzed using Pearson’s Chi-square test, Fisher’s exact test, or Mann–Whitney *U* test, as appropriate. A *p*-value of <0.05 was considered statistically significant.

## 3. Results

### 3.1. Baseline Characteristics of the Participants

Thirty-two out of the forty recruited patients, participated in the study from January 2017 to October 2018. Figure 1 shows a CONSORT flow diagram of this study. Table 1 shows the baseline characteristics of the 32 patients. The clinicopathological characteristics and surgery-related factors, which include operation time, blood loss, and frequency of postoperative complications, did not differ significantly between the two groups.

### 3.2. Primary Endpoint Comparisons between the CT and Placebo Groups on Physical Activity

Regarding exercise and the number of steps, there was no statistically significant difference (*p* > 0.05) between the two groups during the observation period although the CT group showed better performance in every investigated period except on day 3 regarding the number of steps (Figure 2).

### 3.3. Secondary Endpoint Comparisons between the CT and Placebo Groups on Inflammation Indicators

Regarding the indicators of inflammation, white blood cell count (POD 2 and 10), neutrophil (POD 2, 7, and 10), and CRP (POD 13) in the CT group were significantly lower (*p* < 0.05) than that found in the placebo group (Figure 3). There was no statistically significant difference (*p* > 0.05) between the two groups in terms of nutritional indicators, such as serum albumin, prealbumin, RBP, and CONUT score (Figure 4). Additionally, 6MWT on POD 7 in the CT group was significantly higher (*p* < 0.05) than that in the placebo group (Figure 5). Regarding QoR-40, the total score and the subscale scores of Physical Independence and Patient Support were significantly higher in the CT group than in the placebo group on POD 13 (Figure 6).

## 4. Discussion

The main goal of this randomized clinical trial (RCT) was to determine whether perioperative oral administration of CT can promote recovery after esophagectomy among patients with esophageal cancer. Although our study was able to reveal a significant reduction in the inflammatory response after esophagectomy, no statistically significant differences were found in physical activity and various nutritional indicators between the two groups.

A previous RCT regarding CT administration in humans demonstrated that influenza vaccination of poorly nourished elderly people led to higher seroconversion rates in the CT group than in the placebo group [29]. Another RCT showed that the incidence of subjects with colds was significantly lower in the CT group than in the placebo group [30]. These studies suggested that the coadministration of CT could improve immune response. Moreover, another RCT suggested that CT ingestion prevented changes in the inflammatory responses, such as neutrophil count and CRP level, and prevented lowering of the immune state such as lymphocyte after intense endurance exercise [31,32]. With regard to surgical invasion, Miyachi et al. reported that the coadministration of CT led to early recovery from elevated neutrophil counts, interleukin-6, CRP, and body temperature as well as early recovery from decreased lymphocyte count after gastrectomy [10]. We attribute the parity found in the physical activity status of the two groups to a large variation in the exercise performed and the small number of patients recruited. Additionally, it should have been necessary to observe physical activity and nutritional indicators for a longer period after surgery.

To the best of our knowledge, this is the first report with an indication that the oral administration of CT could improve exercise tolerance and QOL during perioperative care in patients who underwent esophagectomy. In this study, patients in the CT group were found to be more tolerant to exercise than their control counterparts. Indeed, exercise tolerance is reported to be associated with hepato–pancreato–biliary and colorectal postoperative complications [33,34], which underscores its further evaluation in patients who underwent esophagectomy in our study. This finding implies that patients managed with oral administration of CT during perioperative care can commence early rehabilitation program to hasten functional recovery. Moreover, in recent years, the evaluation of patients regarding their QOL after surgery has been largely subjective.

Our study demonstrated that patients in the CT group had better overall recovery, particularly in the areas of Physical Independence and Patient Support. This thus implies that CT administration supports early recovery after surgery for esophageal cancer in this study; hence, they can be incorporated into the postoperative early recovery program without burden, while greatly contributing to the perioperative management of esophageal cancer. A low QoR-40 on POD 3 was associated with longer duration of surgery, respiratory complications, and increased length of hospital stay, while QoR-40 on POD 3 was found to be correlated with improved QOL at 3 months after surgery in cardiac surgical patients [24]. Of course, reducing complications is crucial, but the perioperative administration of CT may help patients in early recovery after surgery and enable them to return to daily life and society.

There were no adverse events due to the administration of CT. Only one patient had massive bleeding in the CT group, which was caused by damage to the right bronchial artery when passing the gastric tube through the thoracic cavity. We believe that it was caused by the surgical procedure and cannot be related to the administration of CT.

The small sample of patients and short follow-up duration are the main limitations of this study, which call for caution in utilizing the findings. Nonetheless, the findings have provided insight into the effects of oral CT administration during perioperative care for esophagectomy.

## 5. Conclusions

Based on the outcome of this study, we conclude that perioperative oral administration of CT may contribute to the early recovery of patients after esophagectomy and may facilitate early rehabilitation through the suppression of inflammatory responses that often occur after surgery. Our findings also indicate that patients administered with CT during perioperative care can tolerate exercise better and could experience better recovery. These findings provide useful clinical information regarding the perioperative management of patients with esophageal cancer.

## Figures and Tables

**Figure 1 nutrients-14-02319-f001:**
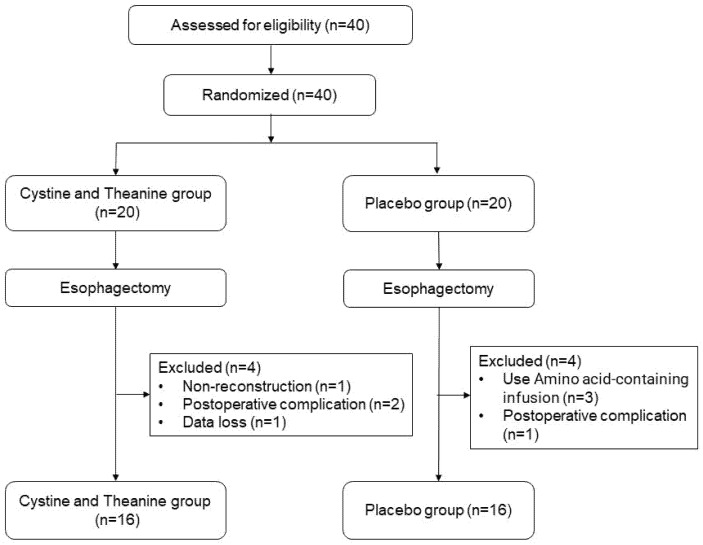
The CONSORT flowchart showing the randomization of patients in the study.

**Figure 2 nutrients-14-02319-f002:**
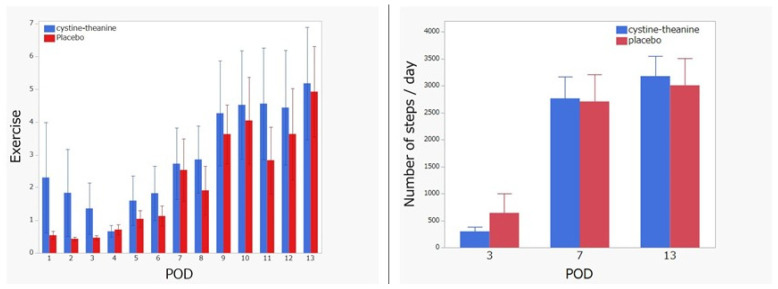
Exercise and number of steps. There was no difference between the two groups, although the CT group showed better exercise and the number of steps in every investigated period (except day 3). Exercise is the value obtained by multiplying metabolic equivalent (MET) and the activity time (h) when the physical activity intensity is ≥3 METs.

**Figure 3 nutrients-14-02319-f003:**
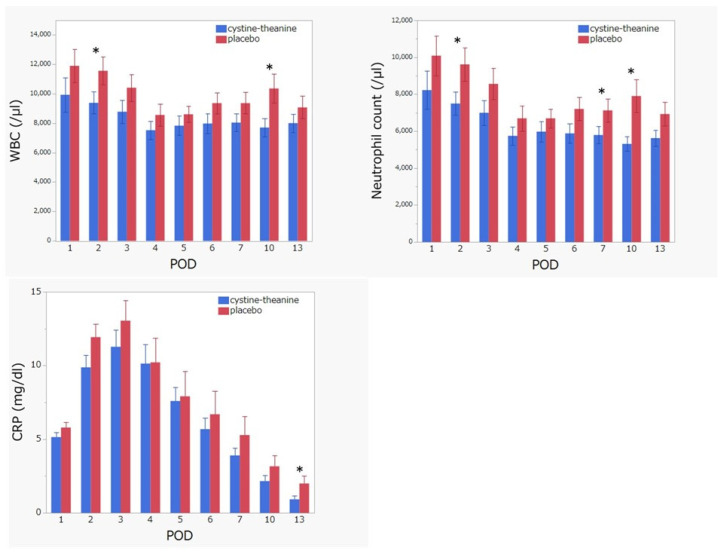
White blood cell count, neutrophil count, and C-reactive protein (CRP). White blood cell count (POD 2 and 10), neutrophil (POD 2, 7, and 10), and CRP (POD 13) in the CT group were significantly lower than in the placebo group. * *p* < 0.05.

**Figure 4 nutrients-14-02319-f004:**
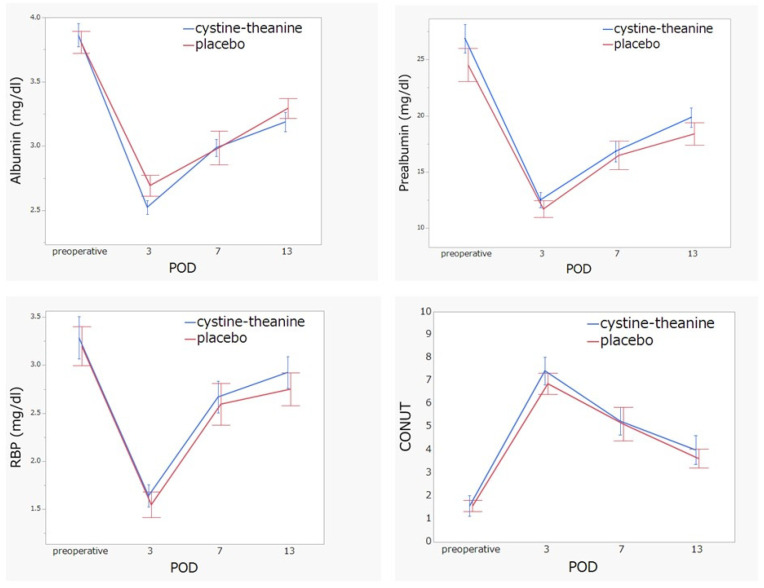
Albumin, prealbumin, retinol-binding protein (RBP), and CONUT score. There were no differences between the two groups in terms of the serum albumin, prealbumin, and RBP levels as well as the CONUT score.

**Figure 5 nutrients-14-02319-f005:**
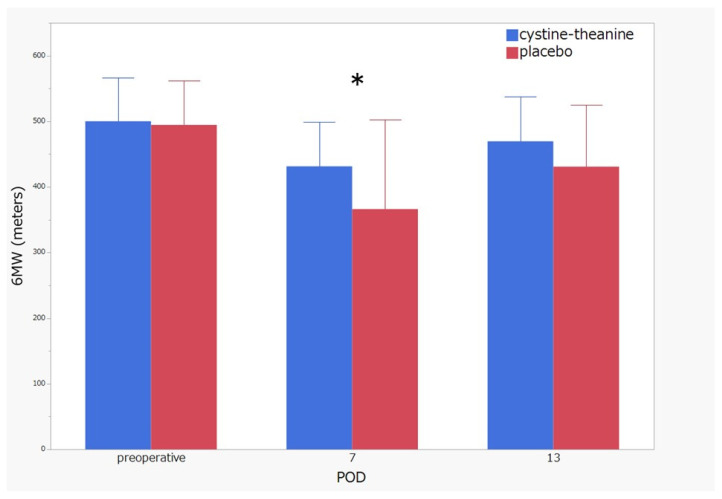
A 6 min walk test (6MWT). 6MWT on POD 7 in the CT group was significantly higher than that in the placebo group. * *p* < 0.05.

**Figure 6 nutrients-14-02319-f006:**
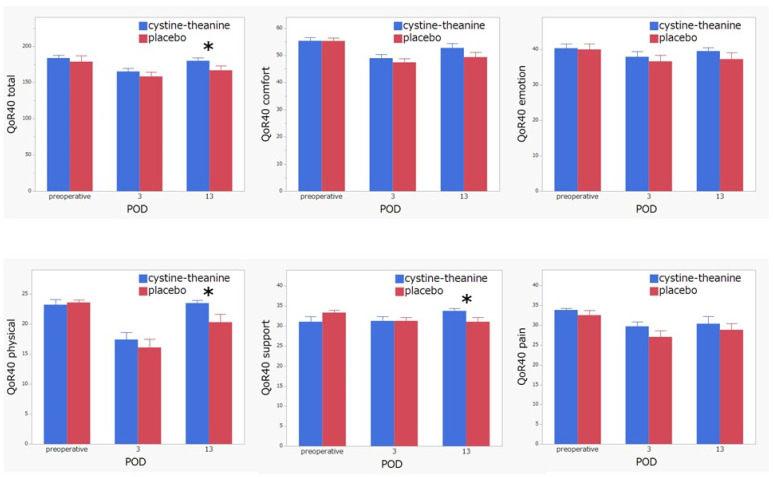
QoR-40. Total score, physical independence, and patient support on POD 13 in the CT group were significantly higher than those in the placebo group. * *p* < 0.05.

**Table 1 nutrients-14-02319-t001:** Comparison of the baseline characteristics of the patients in the CT and placebo groups.

Characteristics	CT * Group (*n* = 16) (%)	P * Group (*n* = 16) (%)	*p*-Value
Age (years)			0.74
Mean ± SE *	68.4 ± 1.4	69.1 ± 1.4	
(Range)	(58–78)	(52–78)	
Gender			0.37
Male	14 (87.5)	12 (75.0)	
Female	2 (12.5)	4 (25.0)	
Location			0.051
Upper third	1 (6.3)	3 (18.8)	
Middle third	8 (50.0)	7 (43.8)	
Lower third	7 (43.8)	2 (12.5)	
Abdominal	0 (0.0)	4 (25.0)	
Histological type			0.69
Squamous cell carcinoma	13 (81.3)	12 (75.0)	
Adenocarcinoma	1 (6.3)	3 (18.8)	
Other	1 (6.3)	1 (6.3)	
Clinical T			0.65
T1	4 (25.0)	6 (37.5)	
T2	4 (25.0)	2 (12.5)	
T3	8 (50.0)	7 (43.8)	
T4a	0 (0.0)	1 (6.3)	
Clinical N			0.43
N0	5 (31.3)	3 (18.8)	
N1	7 (43.8)	11 (68.8)	
N2	4 (25.0)	2 (12.5)	
Clinical Stage			1
Stage I	4 (25.0)	5 (31.3)	
Stage II	5 (31.3)	4 (25.0)	
Stage III	7 (43.8)	6 (37.5)	
Stage VI	0 (0.0)	1 (6.3)	
Pretreatment			0.53
None	3 (18.8)	5 (31.3)	
Chemotherapy	10 (62.5)	10 (62.5)	
Chemoradiotherapy	3 (18.8)	1 (6.3)	
Operation time (min)			0.53
Mean ± SE	665 ± 19.0	682 ± 19.0	
(Range)	(529–815)	(541–777)	
Blood loss (ml)			0.074
Mean ± SE	463 ± 107	182 ± 107	
(Range)	(16–2509)	(23–458)	
Complication (≥Grade 2)			
Pneumonia	2 (12.5)	4 (25.0)	0.37
Anastomosis leakage	2 (12.5)	1 (6.3)	0.54
Colitis	0 (0.0)	2 (12.5)	0.14
Arrhythmia	2 (12.5)	0 (0.0)	0.14
Pyothorax	0 (0.0)	1 (6.3)	0.31
Chylothorax	0 (0.0)	1 (6.3)	0.31

* CT: cystine and theanine, P: placebo, SE: standard error.

## Data Availability

The data used in this study are available from the corresponding author. The data are not publicly available because of ethical restrictions.

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
