# Peer review of "Perioperative Administration of Cystine and Theanine Suppresses Inflammation and Facilitates Early Rehabilitation and Recovery after Esophagectomy: A Randomized, Double-Blind, Controlled Clinical Trial"

_nutrients, 2022, doi:10.3390/nu14112319_

Round 1

Reviewer 1 Report

The article is well write. The statistical anlysis very correct. What can be pointed out is that the selection of patients is so strict that the sixteen patients recruited are very ideal for drawing conclusions. Two data which deserve observation and which should be justified are: blood loss and anastomotic laeakage which are greater, even if not statistically significant, in the CT group.

Author Response

In response to the Reviewer 1’s comments:
We are extremely grateful for your critical review and helpful comments. We have revised the manuscript according to the comments.

The article is well write. The statistical anlysis very correct. What can be pointed out is that the selection of patients is so strict that the sixteen patients recruited are very ideal for drawing conclusions. Two data which deserve observation and which should be justified are: blood loss and anastomotic laeakage which are greater, even if not statistically significant, in the CT group.

Response: Thank you for your comments.

Only one patient had massive bleeding in the CT group, which was caused by damage to the right bronchial artery when passing the gastric tube through the thoracic cavity. We believe that it was caused by the surgical procedure and cannot be related to the administration of CT. We have added this issue to the Discussion section (page 10, lines 271–274).

As the number of anastomotic leakage cases was extremely small in the two groups, we believe that there was no issue.

Reviewer 2 Report

The manuscript entitled “Perioperative administration of cystine and theanine suppresses inflammation and facilitates early rehabilitation and recovery after esophagectomy: a randomized, double-blind, controlled clinical trial” by Okamoto et al. is well structured and clearly written. The limitation of this study is in the small number of patients, a limitation that is however made explicit by the authors. The authors especially in the discussion should talk about the issue of blood loss that seems more consistent in treated patients than in placebos although it is not significant and should make assumptions and warn about this occurrence. Minor reviews are as follows:

Introduction: the studies in references 7 and 8 are not recent

Results: In table 1 the values in the column of p values are misaligned; you should indicate in bold the characteristics (gender, location... up to complication), you should write in full cT and cN

In the legend in figure 2 indicate the meaning of exercise (even if described in methods)

Author Response

In response to the Reviewer 2’s comments:

We are extremely grateful for your critical review and helpful comments. We have revised the manuscript according to the comments.

The manuscript entitled “Perioperative administration of cystine and theanine suppresses inflammation and facilitates early rehabilitation and recovery after esophagectomy: a randomized, double-blind, controlled clinical trial” by Okamoto et al. is well structured and clearly written. The limitation of this study is in the small number of patients, a limitation that is however made explicit by the authors. The authors especially in the discussion should talk about the issue of blood loss that seems more consistent in treated patients than in placebos although it is not significant and should make assumptions and warn about this occurrence. Minor reviews are as follows:

Response: Thank you for your comments.

Only one case had massive bleeding in the CT group, which was caused by damage to the right bronchial artery when passing the gastric tube through the thoracic cavity. We believe that it was caused by the surgical procedure and cannot be related to the administration of CT. We have added this issue to the Discussion section (page 10, lines 271–274).

Introduction: the studies in references 7 and 8 are not recent

Response: We agree with your opinion and have revised “Recent” to “Previous” in the manuscript (page 2, line 61).

Results: In table 1 the values in the column of p values are misaligned; you should indicate in bold the characteristics (gender, location... up to complication), you should write in full cT and cN

Response: We have revised Table 1 as per your suggestions.

In the legend in figure 2 indicate the meaning of exercise (even if described in methods)

Response: We have revised the legend of Figure 2 according to your suggestions.